# Learning Spatiotemporal Dynamical Systems from Point Process Observations

**Valerii Iakovlev**    **Harri Lähdesmäki**
Department of Computer Science, Aalto University, Finland
{valerii.iakovlev, harri.lahdesmaki}@aalto.fi

## Abstract

Spatiotemporal dynamics models are fundamental for various domains, from heat propagation in materials to oceanic and atmospheric flows. However, currently available neural network-based spatiotemporal modeling approaches fall short when faced with data that is collected randomly over time and space, as is often the case with sensor networks in real-world applications like crowdsourced earthquake detection or pollution monitoring. In response, we developed a new method that can effectively learn spatiotemporal dynamics from such point process observations. Our model integrates techniques from neural differential equations, neural point processes, implicit neural representations and amortized variational inference to model both the dynamics of the system and the probabilistic locations and timings of observations. It outperforms existing methods on challenging spatiotemporal datasets by offering substantial improvements in predictive accuracy and computational efficiency, making it a useful tool for modeling and understanding complex dynamical systems observed under realistic, unconstrained conditions.

## 1 Introduction

In this work, consider the modeling of spatiotemporal dynamical systems whose dynamics are driven by partial differential equations. Such systems are ubiquitous and range from heat propagation in microstructures to the dynamics of oceanic currents. We use the data-driven modeling approach, where we observe a system at various time points and spatial locations and use the collected data to learn a model. In practice, the data is often collected by sensor networks that make measurements at random time points and random spatial locations. Such a measurement approach is used, for example, in crowdsourced earthquake monitoring where smartphones are used as measurement devices (Minson et al., 2015; Kong et al., 2016), in oceanographic monitoring where measurements are made by floating buoys (Albaladejo et al., 2010; Xu et al., 2014; Marin-Perianu et al., 2008), and for air pollution monitoring with vehicle-mounted sensors (Ma et al., 2008; Ghanem et al., 2004). This approach offers several advantages as it needs no sensor synchronization and allows the sensors to move freely. However, its random nature makes modeling more challenging as it requires capturing both the system dynamics and the random observation process.

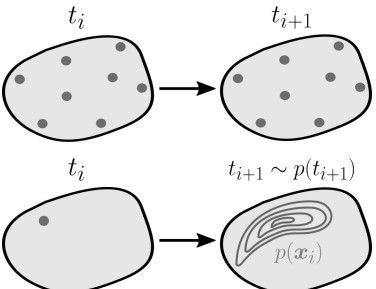

Figure 1: **Top:** Previous models assume dense observations at every time point and fixed spatiotemporal grids. **Bottom:** Our model works with extremely sparse observations and predicts where the next observations will happen.

We focus on the marked spatiotemporal point process setting, where the randomness in observation times and locations is defined in terms of the state of the underlying spatiotemporal dynamical system and marks are generated from the system state via an observation function. To the best of our knowledge, existing neural network-based methods cannot model such randomly observed spatiotemporal dynamical systems, mainly for two reasons. First, they do not model observation times and locations, hence they cannot predict where and when the next observation will be made.

---

Source code and datasets can be found in our github repository.

Second, they assume the sensors form a fixed spatial grid and record data simultaneously, which is not the case in our setup where the data could come from as little as a single sensor at each time point (Fig. 1). For example, some earlier methods assume the observations are made on a fixed and regular spatiotemporal grid (Long et al., 2018; Geneva & Zabaras, 2020), other methods work with irregular, but still fixed, observation locations (Iakovlev et al., 2021; Lienen & Günnemann, 2022), and other works go further and allow the observation locations to change over time (Pfaff et al., 2021; Yin et al., 2023) but fix the observation times and assume dense observations. Whereas, another line of research has proposed methods that model only the spatiotemporal observation process without modeling the system dynamics (Chen et al., 2021; Zhu et al., 2021; Zhou et al., 2022; Zhou & Yu, 2023; Du et al., 2016).

Our work fills this gap and proposes a model for randomly observed spatiotemporal dynamical systems. Our model incorporates techniques from amortized variational inference (Kingma & Welling, 2013), neural differential equations (Chen et al., 2018; Rackauckas et al., 2020), neural point processes (Mei & Eisner, 2017; Chen et al., 2021), and implicit neural representations (Chen et al., 2023; Yin et al., 2023) to efficiently learn both the underlying system dynamics and the random observation process. Our model uses initial observations to obtain the variational estimate of the latent initial state via a transformer encoder (Vaswani et al., 2017), simulates the latent trajectory with neural ODEs (Chen et al., 2018), and uses implicit neural representations to parameterize the point process and observation distribution. Furthermore, we identify a computational bottleneck in the latent state evaluation and propose a technique to alleviate it, resulting in up to 4x faster training. Our model shows strong empirical results outperforming other models from the literature on challenging spatiotemporal datasets.

## 2 BACKGROUND

### 2.1 SPATIOTEMPORAL POINT PROCESSES

Spatiotemporal point processes (STPP) model sequences of events occurring in space and time. Each event has an associated event time $t_i \in \mathbb{R}_{\geq 0}$ and event location $\boldsymbol{x}_i \in \mathbb{R}^{d_{\boldsymbol{x}}}$. Given an event history $\mathcal{H}_t \triangleq \{(t_i, \boldsymbol{x}_i) \mid t_i < t\}$ with all events up to time $t$, we can characterize an STPP by its conditional intensity function

$$\lambda^*(t, \boldsymbol{x}) \triangleq \lim_{\delta t \downarrow 0, \, \delta r \downarrow 0} \frac{\mathbb{P}(t_i \in [t, t + \delta t], \boldsymbol{x}_i \in B_{\delta r}(\boldsymbol{x}) | \mathcal{H}_t)}{\delta t |B_{\delta r}(\boldsymbol{x})|}, \tag{1}$$

where $\delta t$ denotes an infinitesimal time interval, and $B_{\delta r}(\boldsymbol{x})$ denotes a $\delta r$-ball centered at $\boldsymbol{x}$. Given a history $\mathcal{H}_t$ with $i - 1$ events, $\lambda^*(t, \boldsymbol{x})$ describes the instantaneous probability of the next, $i$th, event occurring at time $t$ and location $\boldsymbol{x}$. Given a sequence of $N$ events $\{(t_i, \boldsymbol{x}_i)\}_{i=1}^N$ on a bounded domain $A \subset [0, T] \times \mathbb{R}^{d_{\boldsymbol{x}}}$, the log-likelihood for the STPP is evaluated as (Daley et al., 2003)

$$\log p(\{(t_i, \boldsymbol{x}_i)\}_{i=1}^N) = \sum_{i=1}^N \log \lambda^*(t_i, \boldsymbol{x}_i) - \int_A \lambda^*(t, \boldsymbol{x}) d\boldsymbol{x} dt. \tag{2}$$

Marked STPP extends the above simple STPP by a mark $\boldsymbol{y}_i \in \mathbb{R}^{d_{\boldsymbol{y}}}$ that is associated to each event $(t_i, \boldsymbol{x}_i)$.

### 2.2 ORDINARY AND PARTIAL DIFFERENTIAL EQUATIONS

Given a deterministic continuous-time dynamic system with state $\boldsymbol{z}(t) \in \mathbb{R}^{d_{\boldsymbol{z}}}$, we can describe the evolution of its state in terms of an ordinary differential equation (ODE)

$$\frac{d\boldsymbol{z}(t)}{dt} = f(t, \boldsymbol{z}(t)). \tag{3}$$

For an initial state $\boldsymbol{z}_1$ at time $t_1$ we can solve the ODE to obtain the system state $\boldsymbol{z}(t)$ at later times $t > t_1$. The solution exists and is unique if $f$ is continuous in time and Lipschitz continuous in state (Coddington et al., 1956), and can be obtained either analytically or using numerical ODE solvers (Heirer et al., 1987). In this work we solve ODEs numerically using ODE solvers from

`torchdiffeq` (Chen, 2018) package. Similarly, the dynamics of spatiotemporal systems with state $\boldsymbol{z}(t, \boldsymbol{x})$ defined over both space and time is described in terms of a partial differential equation (PDE)

$$\frac{\partial \boldsymbol{z}(t, \boldsymbol{x})}{\partial t} = F(t, \boldsymbol{x}, \boldsymbol{z}(t, \boldsymbol{x}), \nabla \boldsymbol{z}(t, \boldsymbol{x})) \tag{4}$$

which incorporates both temporal and spatial derivatives, indicated by $\frac{\partial}{\partial t}$ and $\nabla$, respectively.

## 3 PROBLEM SETUP

In this work, we model spatiotemporal dynamical systems from data. The data consists of multiple trajectories collected by observing a system over a period of time. A trajectory consists of $N$ triplets $\{(t_i, \boldsymbol{x}_i, \boldsymbol{y}_i)\}_{i=1}^N$, where $\boldsymbol{y}_i \in \mathbb{R}^{d_y}$ is a system observation, with $t_i \in \mathbb{R}_{\geq 0}$ and $\boldsymbol{x}_i \in \mathbb{R}^{d_x}$ being the corresponding observation time and location. The number of observations $N$ can change across different trajectories in the dataset. Due to randomness of the observation process we assume the observation times and locations do not overlap (neither within a trajectory nor across trajectories), resulting in a single observation $\boldsymbol{y}_i$ per time point and location. For brevity, we describe our method for a single observed trajectory, but extension to multiple trajectories is straightforward.

We assume the data generating process (DGP) consists of a latent spatiotemporal state $\boldsymbol{u}(t, \boldsymbol{x}) \in \mathbb{R}^{d_u}$ with $t \in \mathbb{R}_{\geq 0}$ and $\boldsymbol{x} \in \mathbb{R}^{d_x}$, whose dynamics are governed by a PDE. The latent state directly affects the times and locations of the observations, and is only partially observed:

$$\boldsymbol{u}(t_1, \cdot) \sim p(\boldsymbol{u}), \quad \frac{\partial \boldsymbol{u}(t, \boldsymbol{x})}{\partial t} = F(\boldsymbol{u}(t, \boldsymbol{x}), \nabla \boldsymbol{u}(t, \boldsymbol{x}))), \tag{5}$$

$$t_i, \boldsymbol{x}_i \sim \mathrm{nhpp}(\lambda(\boldsymbol{u}(t, \boldsymbol{x}))), \qquad i = 1, \ldots, N, \tag{6}$$

$$\boldsymbol{y}_i \sim p(\boldsymbol{y}_i | g(\boldsymbol{u}(t_i, \boldsymbol{x}_i))), \qquad i = 1, \ldots, N. \tag{7}$$

According to this process, each trajectory is generated as follows. We first sample the latent field $\boldsymbol{u}(t_1, \cdot)$ at initial time point $t_1$ and specify its dynamics by a PDE (Eq. 5). Next, we sample the observation times $t_i$ and locations $\boldsymbol{x}_i$ from a non-homogeneous Poisson process (nhpp) with intensity $\lambda$ that is a function of the latent state (Eq. 6). Finally, we sample $\boldsymbol{y}_i$ from the observation distribution parameterized by a function $g$ (Eq. 7). All datasets in this work were generated according to this process (see Appendix A for details). We assume the data generating process is fully unknown, and our goal is to construct and learn its model from the data.

## 4 METHODS

In this section we describe our proposed model, its components, and the parameter inference method.

### 4.1 MODEL

Our goal is to model the true DGP (Eqs. 5-7). To this end, we define our generative model as:

$$\boldsymbol{z}(t_1) \sim p(\boldsymbol{z}(t_1)), \quad \frac{d\boldsymbol{z}(\tau)}{dt} = f(\boldsymbol{z}(\tau)), \tag{8}$$

$$\boldsymbol{u}(t, x) = \phi(\boldsymbol{z}(t), \boldsymbol{x}), \tag{9}$$

$$t_i, \boldsymbol{x}_i \sim \mathrm{nhpp}(\lambda(\boldsymbol{u}(t, \boldsymbol{x}))), \qquad i = 1, \ldots, N, \tag{10}$$

$$\boldsymbol{y}_i \sim p(\boldsymbol{y}_i | g(\boldsymbol{u}(t_i, \boldsymbol{x}_i))), \qquad i = 1, \ldots, N, \tag{11}$$

and the corresponding joint distribution is (details in App. B):

$$p(\boldsymbol{z}_1, \{t_i, \boldsymbol{x}_i, \boldsymbol{y}_i\}_{i=1}^N) = p(\boldsymbol{z}_1) p(\{t_i, \boldsymbol{x}_i\}_{i=1}^N | \boldsymbol{z}_1) \prod_{i=1}^N p(\boldsymbol{y}_i | \boldsymbol{z}_1, t_i, \boldsymbol{x}_i). \tag{12}$$

Below, we describe each component in detail, and a diagram of our model is shown in Figure 2.

**Latent dynamics (Eq. 8).** Our goal is to model the latent PDE dynamics of the DGP (Eq. 5). To do that, we introduce a low-dimensional *temporal* latent state $\boldsymbol{z}(\tau) \in \mathbb{R}^{d_z}$ with $\tau \in \mathbb{R}_{\geq 0}$ that encodes the *spatiotemporal* field $\boldsymbol{u}(\tau, \cdot)$, and introduce a neural network-parameterized ODE governing its dynamics (Eq. 8). We use a low-dimensional latent state because it allows to simulate the dynamics considerably faster than a full-grid spatiotemporal discretization (Wu et al., 2022; Yin et al., 2023). During our experiments, we observed that it takes ODE solvers a long time to solve the ODE and that this time increases with the number of observations. We found this happens because of a bottleneck in existing ODE solvers caused by extremely dense time grids (such as in Fig. 3). Such time grids force the solver to choose small steps which prevents it from adaptively selecting the optimal step size thus reducing its efficiency. To alleviate this bottleneck, we move away from the original dense time grid $t_1, \ldots, t_N$ to an auxiliary sparse grid $\tau_1, \ldots, \tau_n$, with $n \ll N$, where the first and last time points coincide with the original grid: $\tau_1 = t_1, \tau_n = t_N$ (see Fig. 2). Then, we solve for the latent state $\boldsymbol{z}(t)$ only at the sparse grid, which allows the ODE solver to choose the optimal step size and results in approximately an order of magnitude (up to 9x) faster simulations (see Sec. 5). Finally, we use the computed latent states $\boldsymbol{z}(\tau_1), \ldots, \boldsymbol{z}(\tau_n)$ to approximate the original latent state $\boldsymbol{z}(t)$ at intermediate time points via interpolation: $\tilde{\boldsymbol{z}}(t) = \text{interpolate}(t; \boldsymbol{z}(\tau_1), \ldots, \boldsymbol{z}(\tau_n))$.

**Latent state decoding (Eq. 9).** Next, we need to recover the latent spatiotemporal state $\boldsymbol{u}(t, \boldsymbol{x})$ from the low-dimensional representation $\tilde{\boldsymbol{z}}(t)$. We define the latent spatiotemporal state $\boldsymbol{u}(t, \boldsymbol{x})$ as a function $\phi(\tilde{\boldsymbol{z}}(t), \boldsymbol{x})$ of the latent state $\tilde{\boldsymbol{z}}(t)$ and evaluation location $\boldsymbol{x}$ (Eq. 9). In particular, we parameterize $\phi(\tilde{\boldsymbol{z}}(t), \boldsymbol{x})$ by an MLP which takes $\tilde{\boldsymbol{z}}(t) + \text{proj}(\boldsymbol{x})$ as the input and maps it directly to $\boldsymbol{u}(t, \boldsymbol{x})$, where $\text{proj}(\boldsymbol{x}) \in \mathbb{R}^{d_z}$ is a trainable linear mapping of $\boldsymbol{x}$ to $\mathbb{R}^{d_z}$. This formulation results in space-time continuous $\boldsymbol{u}(t, \boldsymbol{x})$ which can be evaluated at any time point and spatial location, which is required to define the intensity function $\lambda(\boldsymbol{u}(t, \boldsymbol{x}))$ and observation function $g(\boldsymbol{u}(t, \boldsymbol{x}))$.

**Intensity function (Eq. 10).** We parameterize the intensity function $\lambda(\boldsymbol{u}(t, \boldsymbol{x}))$ by an MLP which takes $\boldsymbol{u}(t, \boldsymbol{x})$ as the input and maps it to intensity of the Poisson process (Eq. 10). To ensure non-negative values of the intensity function and improve its numerical stability we further exponentiate the output of the MLP and add a small constant to it.

**Observation function (Eq. 11).** The observation function $g(\boldsymbol{u}(t, \boldsymbol{x}))$ maps the latent spatiotemporal state $\boldsymbol{u}(t, \boldsymbol{x})$ to parameters of the observation model (Eq. 11). Therefore, its exact structure depends on the observation model. In this work the observation model is a normal distribution, where the variance is fixed and the observation function $g(\boldsymbol{u}(t, \boldsymbol{x}))$ returns the mean.

Details of the model specification and parameterization can be found in Appendix B and C.

## 4.2 PARAMETER AND LATENT STATE INFERENCE

To infer the model parameters and posterior of the latent initial state $\boldsymbol{z}_1$ we use variational inference (Blei et al., 2017). We define an approximate posterior $q(\boldsymbol{z}_1; \psi)$ with variational parameters $\psi$ to approximate the true posterior $p(\boldsymbol{z}_1 | \{t_i, \boldsymbol{x}_i, \boldsymbol{y}_i\}_{i=1}^{N})$ and then minimize the Kullback-Leibler divergence

$$\text{KL}\left[q(\boldsymbol{z}_1; \psi) \| p(\boldsymbol{z}_1 | \{t_i, \boldsymbol{x}_i, \boldsymbol{y}_i\}_{i=1}^{N})\right] \tag{13}$$

over the model and variational parameters to obtain an estimate of the model parameters and approximation of the posterior.

To avoid optimizing the local variational parameters $\psi$ for each trajectory in the dataset, we use amortization (Kingma & Welling, 2013) and define $\psi$ as the output of an encoder:

$$\psi = \text{Encoder}(\{t_i, \boldsymbol{x}_i, \boldsymbol{y}_i\}_{i=1}^{N}), \tag{14}$$

which maps observations $\{t_i, \boldsymbol{x}_i, \boldsymbol{y}_i\}_{i=1}^{N}$ to the local variational parameters $\psi$. This allows to optimize a fixed set of encoder parameters instead of a dataset size-dependent set of local variational parameters. We discuss the structure of the encoder in the next section.

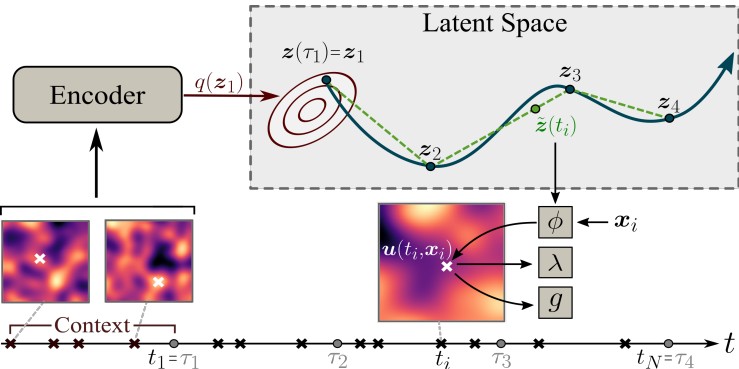

**Figure 2: Model diagram**. Black crosses on the time axis are observation times, while white crosses on field images are the corresponding observation locations. The initial observations (context) are mapped by the encoder to the latent initial state distribution $q(\boldsymbol{z}_1)$. The initial state $\boldsymbol{z}_1$ is then sampled from that distribution and evolved through time using the dynamics model. The latent trajectory is evaluated only at the sparse time grid $\tau_1, \ldots, \tau_n$ and the latent state $\tilde{\boldsymbol{z}}(t)$ at other time points is evaluated via interpolation. The latent state $\tilde{\boldsymbol{z}}(t)$ is then mapped by $\phi$ to the spatiotemporal state $\boldsymbol{u}(t, \boldsymbol{x})$, which is used to parameterize the point process and observation distribution (via mappings $\lambda$ and $g$, respectively) for predicting subsequent observation times and locations.

In practice, instead of minimizing the KL divergence we, equivalently, maximize the evidence lower bound (ELBO) which for our model is defined as:

$$\mathcal{L} = \underbrace{\sum_{i=1}^{N} \mathbb{E}_{q(\boldsymbol{z}_1;\psi)}\left[\ln p(\boldsymbol{y}_i|t_i, \boldsymbol{x}_i, \boldsymbol{z}_1)\right]}_{\text{(i) Expected observation log-lik.}} + \underbrace{\mathbb{E}_{q(\boldsymbol{z}_1;\psi)}\left[\sum_{i=1}^{N} \ln \lambda(\boldsymbol{u}(t_i, \boldsymbol{x}_i)) - \int \lambda(\boldsymbol{u}(t, \boldsymbol{x}))d\boldsymbol{x}dt\right]}_{\text{(ii) Expected STPP log-lik.}} \quad (15)$$

$$- \underbrace{\text{KL}[q(\boldsymbol{z}_1;\psi)\|p(\boldsymbol{z}_1)]}_{\text{(iii) KL between prior and posterior}}. \quad (16)$$

The ELBO is maximized wrt. the model and encoder parameters. Appendix B contains detailed derivation of the ELBO, and fully specifies the model and the approximate posterior. While the term *(iii)* can be computed analytically, computation of terms *(i)* and *(ii)* involves approximations: Monte Carlo integration for the expectations and intensity integral, and numerical ODE solvers for the solution of the initial value problems. Appendix B details the computation of ELBO.

## 4.3 ENCODER

Our encoder maps the observations $\{t_i, \boldsymbol{x}_i, \boldsymbol{y}_i\}_{i=1}^{N}$ to the local variational parameters $\psi$ of $q(\boldsymbol{z}_1;\psi)$ using Eq. 14. This process begins by embedding each observation into a high-dimensional feature space. The embedded sequence is then processed by a stack of transformer encoder layers (Vaswani et al., 2017), after which the output is mapped to the variational parameters $\psi$. Below, we describe this process in more detail.

First, we convert the observation sequence $\{t_i, \boldsymbol{x}_i, \boldsymbol{y}_i\}_{i=1}^{N}$ into a sequence of vectors $\{\boldsymbol{b}_i\}_{i=1}^{N}$, where

$$\boldsymbol{b}_i = \text{proj}(t_i) + \text{proj}(\boldsymbol{x}_i) + \text{proj}(\boldsymbol{y}_i), \quad (17)$$

and proj are separate trainable linear projections. This additive decomposition is simple yet shows strong empirically performance in our experiments. Since $t_i$ and $\boldsymbol{x}_i$ already provide positional information, we omit positional encodings. Additionally, we introduce an aggregation token, [AGG], with a learnable representation to indicate where the encoder should aggregate information about the observations. This gives us the following encoder input sequence: $\{\{\boldsymbol{b}_i\}_{i=1}^{N}, [\text{AGG}]\}$. Next, a stack of transformer encoder layers processes the input sequence, producing the output sequence $\{\{\overline{\boldsymbol{b}}_i\}_{i=1}^{N}, \overline{[\text{AGG}]}\}$. Finally, we map the output aggregation token $\overline{[\text{AGG}]}$ to the variational parameters $\psi$. This mapping depends on the form of the approximate posterior (see Appendix B for specification of the mapping).

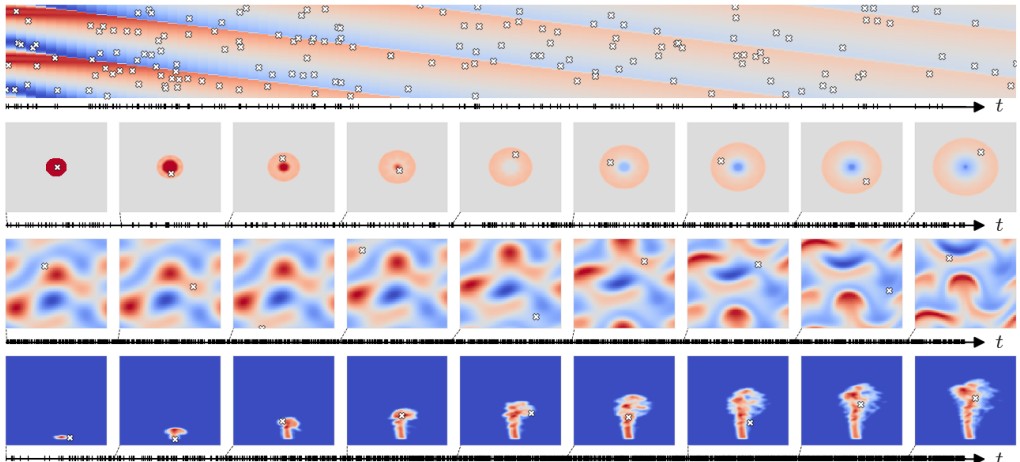

Figure 3: **Examples of trajectories from our datasets.** White crosses indicate the observation locations $\boldsymbol{x}$, marks on the horizontal time grid indicate the corresponding observation times $t$, and color-coded fields denote the system state with observations $\boldsymbol{y}$ at white crosses $(t, \boldsymbol{x})$. **Row 1:** Burger's (1D) dataset. The full trajectory is shown, with 1-D measurement location $\boldsymbol{x}$ along the vertical and time $t$ along the horizontal directions. **Rows 2, 3 and 4:** Shallow Water, Navier-Stokes, and Scalar Flow datasets, respectively. Due to the 2D nature of the systems, only snapshots of the system state are plotted.

## 4.4 FORECASTING

Given a set of initial observations $\{t_i^*, \boldsymbol{x}_i^*, \boldsymbol{y}_i^*\}_{i=1}^{N_{\text{ctx}}}$ of a test trajectory (we call it "context"), we want to predict how the system behaves at some time point $t > t_{N_{\text{ctx}}}$ and spatial location $\boldsymbol{x}$. Given the context, we predict the observation $\boldsymbol{y}$ at $t$ and $\boldsymbol{x}$ as the expectation of the following approximate posterior predictive distribution:

$$p(\boldsymbol{y}|t, \boldsymbol{x}, \{t_i^*, \boldsymbol{x}_i^*, \boldsymbol{y}_i^*\}_{i=1}^{N_{\text{ctx}}}) = \int p(\boldsymbol{y}|t, \boldsymbol{x}, \boldsymbol{z}_1) q_{\psi^*}(\boldsymbol{z}_1) d\boldsymbol{z}_1, \tag{18}$$

where $\psi^* = \text{Encoder}(\{t_i^*, \boldsymbol{x}_i^*, \boldsymbol{y}_i^*\}_{i=1}^{N_{\text{ctx}}})$, and the expectation is approximated via Monte Carlo integration. In Eq. 18, we omit explicitly conditioning on training data for brevity. Similarly, the distribution over the observation times and locations is evaluated as:

$$p(t, \boldsymbol{x}|\{t_i^*, \boldsymbol{x}_i^*, \boldsymbol{y}_i^*\}_{i=1}^{N_{\text{ctx}}}) = \int p(t, \boldsymbol{x}|\boldsymbol{z}_1) q_{\psi^*}(\boldsymbol{z}_1) d\boldsymbol{z}_1. \tag{19}$$

In our experiments, the context for each trajectory is defined as all observations withing a fixed initial time period (see Appendix A for details).

## 5 EXPERIMENTS

In this section we demonstrate properties of our method and compare it against other methods from the literature. Our datasets are generated by three commonly-used PDE systems: Burgers' (models nonlinear 1D wave propagation), Shallow Water (models 2D wave propagation under the gravity), and Navier-Stokes with transport (models the spread of a pollutant in a liquid over a 2D domain). In addition to the synthetic data, we include a real-world dataset Scalar Flow (Eckert et al., 2019), which contains observations of smoke plumes raising in warm air. See Appendix A for details about the dataset generation. Figure 3 shows examples of trajectories from the datasets.

In all cases our model has at most 3 million parameters, and training takes at most 1.5 hours on a single GeForce RTX 3080 GPU. The training is done for 25k iterations with learning rate 3e-4 and batch size 32. We use the adaptive ODE solver (dopri5) from `torchdiffeq` package with relative

and absolute tolerance set to 1e-5. As performance metrics we use mean absolute error (MAE) for the observations $\boldsymbol{y}_i$, and event-averaged log-likelihood of the point process for $t_i$ and $\boldsymbol{x}_i$, both evaluated on the test set. See Appendix C for details about our training setup.

**Context Size.** Our encoder maps the initial observations (context) $\{t_i^*, \boldsymbol{x}_i^*, \boldsymbol{y}_i^*\}_{i=1}^{N_{\text{ctx}}}$ to parameters $\psi$ of the approximate posterior $q(\boldsymbol{z}_1; \psi)$. Here we look at how accuracy of the state predictions and process likelihood is affected by the context size. We train and test our model with different context sizes: full (using all $N_{\text{ctx}}$ points), half of the context (using points from $N_{\text{ctx}}/2$ to $N_{\text{ctx}}$), etc., and show the results in Figure 4. We see that both MAE and process likelihood improve as we increase the context size, but the improvements tend to saturate for larger contexts. This indicates that the encoder mostly uses observations that are close to the time point at which the latent state is inferred, and does not utilize observations that are too far away from it. This effect is especially visible with synthetic data, but the plots suggest the real-world dataset Scalar Flow could still benefit from larger context.

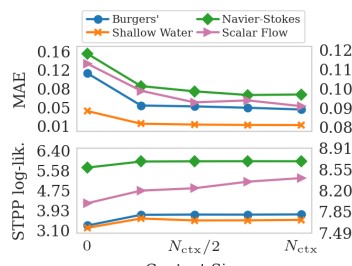

Figure 4: Test MAE ($\downarrow$) and log-likelihood ($\uparrow$) vs. context size. For Scalar Flow we use the right axis for better visibility.

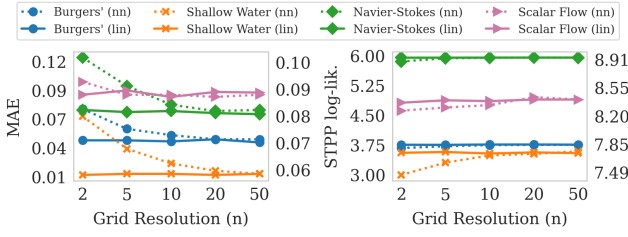

Figure 5: Test MAE ($\downarrow$) and process log-likelihood ($\uparrow$) for different temporal grid resolutions ($n$) and interpolation methods (nn = nearest neighbor, lin = linear). For Scalar Flow we use the right axis for better visibility.

| Dataset | Interp. | Seq. |
|---|---|---|
| Burgers' | 0.007 | 0.018 |
| Shallow Water | 0.011 | 0.045 |
| Navier-Stokes | 0.012 | 0.108 |
| Scalar Flow | 0.011 | 0.089 |

Table 1: Latent state evaluation time (sec), interpolation vs. sequential method. Averaged time over all trajectories.

**Latent State Interpolation.** As discussed in Section 4.1, we do not evaluate the latent state at the full time grid $t_1, \ldots, t_N$ using the ODE solver (we call it the sequential method). Instead, we use an adaptive ODE solver to evaluate the latent state only at a sparser time grid $\tau_1, \ldots, \tau_n$, and then interpolate the evaluations to the full grid. Here we look at how this approach affects training times, and investigate the effects of the grid resolution $n$ and different interpolation methods. We vary the resolution from coarse $n = 2$ to fine $n = 50$, and test nearest neighbor and linear interpolation methods. Figure 5 shows that both MAE and process log-likelihood improve with the grid resolution, but only when we use nearest neighbor interpolation. With linear interpolation, the model achieves its best performance with just $n = 2$, meaning that interpolation is done between only two points in the latent space, and further increasing the resolution does not improve the results. To study this observation in more detail, we provide additional experiments in Appendix D. Both interpolation methods achieve the same optimal performance when $n \geq 20$. This effect is visible for both synthetic and, to a smaller extent, real data. Furthermore, Table 1 shows that our method results in up to 9x faster latent state evaluation than the sequential method, which in our case translates in up to 4x faster training.

**Latent Space Dimension.** Our model assumes the system dynamics can be accurately modeled in a low-dimensional latent space. In this experiment we show that this assumption is valid at least for the four systems that we consider and best predictive accuracy is achieved for relatively low-dimensional latent space dimension $d_{\boldsymbol{z}}$. In Figure 6 we show how MAE and process log-likelihood improve as we increase the latent state dimension. For most datasets the predictive accuracy converges for as low as 16-dimensional latent space, except for the Navier-Stokes dataset, where $d_{\boldsymbol{z}} = 64$ is required, likely

due to its more intricate state (as can be qualitatively observed in Fig. 3). We note that $d_z$ affects the number of model parameters, but the difference between 1- and 64-dimensional latent space is less than 3%.

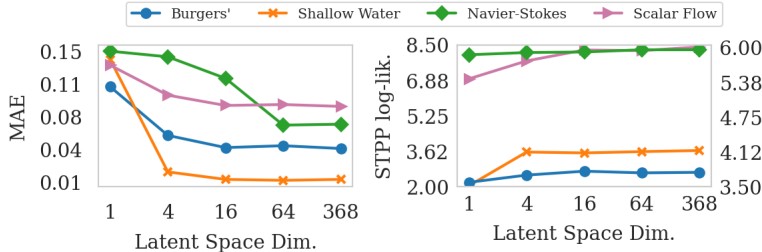

Figure 6: Test MAE ($\downarrow$) and process log-likelihood ($\uparrow$) for different latent space dimensions ($d_z$). For Burgers' and Navier-Stokes we use the right axis for better visibility.

**Interaction Between Observation and Process Models.** In our model the observation times and locations are modeled by a point process model (Eq. 10), while observations are modeled by an observation model (Eq. 11). Intuitively, one could expect some interaction between them and ask the question of how important is one model for the performance of the other. In this experiment we look at the effect of removing the point process model on the performance of the observation model and vice versa. We remove the point process model by removing Eq. 10 from our model, so that we do not model the observation times and locations.

| Dataset | MAE | MAE (-STPP) | Log-lik. | Log-lik (-Obs.) |
|---|---|---|---|---|
| Burgers' | 0.042 | 0.043 | 3.77 | 3.70 |
| Shallow Water | 0.012 | 0.012 | 3.67 | 3.56 |
| Navier-Stokes | 0.071 | 0.071 | 5.96 | 5.92 |
| Scalar Flow | 0.090 | 0.089 | 8.43 | 6.87 |

Table 2: Test MAE ($\downarrow$) and log-likelihood ($\uparrow$) of the full model, and after removing the point process (-STPP) or the observation model (-Obs.).

Similarly, we remove the observation model by removing Eq. 11 from our model and also removing observations $y_i$ from our data, so that there is no information about the observation values. In Table 2 we show that removing the point process model seems to have practically no effect on MAE, while removing the observation model decreases the process likelihood. This shows that if we know the system states we can model the observation times and locations more accurately.

**Comparison to Other Methods.** We compare our method against two groups of methods from the literature. The first group consists of neural spatiotemporal dynamic models that can model only the observations $y_i$ but not the time points and spatial locations. This group includes FEN (Lienen & Günnemann, 2022) (graph neural network-based model, simulates the system dynamics at every point on the spatial grid), NSPDE (Salvi et al., 2022) (space- and time-continuous model simulating the dynamics in the spectral domain), and CNN-ODE (our simple baseline that uses a CNN encoder/decoder to map observations to/from the latent space, and uses neural ODEs (Chen et al., 2018) to model the latent space dynamics). Note that due to their assumption about multiple observations per time point, dynamic models require time binning (see Appendix C.4). The second group consists of neural spatiotemporal point processes that model only the observation times and locations. This group consists of DSTPP (Yuan et al., 2023), NSTPP (Chen et al., 2021), and AutoSTPP (Zhou & Yu, 2023). We also provide simple baselines for both groups: median predictor for dynamic models, and constant (learnable) intensity for the point process group. Hyperparameters of each method were tuned for the best performance. See Appendix C.4 for details about the models and hyperparameters used.

Table 3 shows the comparison results. We see that most methods from the first group perform rather poorly and fail to beat even the MAE of the median predictor, with the CNN-ODE model showing the best results. However, CNN-ODE still performs considerably worse than our model on synthetic data, likely because of the loss of information caused by time binning. On the real-world Scalar Flow dataset CNN-ODE and our model have similar performance. In the second group most methods

| Model | Burgers' | | Shallow Water | | Navier-Stokes | | Scalar Flow | |
|---|---|---|---|---|---|---|---|---|
| | MAE ($\downarrow$) | Log-lik. ($\uparrow$) | MAE ($\downarrow$) | Log-lik. ($\uparrow$) | MAE ($\downarrow$) | Log-lik. ($\uparrow$) | MAE ($\downarrow$) | Log-lik. ($\uparrow$) |
| Median Pred. | 0.161 | - | 0.148 | - | 0.155 | - | 0.140 | - |
| FEN | N/A | - | $0.336 \pm 0.008$ | - | $0.158 \pm 0.006$ | - | $0.197 \pm 0.003$ | - |
| NSPDE | $0.166 \pm 0.002$ | - | $0.339 \pm 0.002$ | - | $0.098 \pm 0.001$ | - | $0.142 \pm 0.003$ | - |
| CNN-ODE | $0.076 \pm 0.000$ | - | $0.032 \pm 0.001$ | - | $0.077 \pm 0.002$ | - | $\mathbf{0.091 \pm 0.001}$ | - |
| Const. Intensity | - | 3.59 | - | 2.02 | - | 5.86 | - | 6.87 |
| DSTPP | - | $0.67 \pm 0.13$ | - | $0.08 \pm 0.04$ | - | $0.41 \pm 0.04$ | - | $3.02 \pm 0.01$ |
| NSTPP | - | $2.35 \pm 0.06$ | - | $1.53 \pm 0.13$ | - | $3.83 \pm 0.14$ | - | $6.69 \pm 0.03$ |
| AutoSTPP | - | N/A | - | $3.37 \pm 0.01$ | - | $5.91 \pm 0.01$ | - | $\mathbf{8.49 \pm 0.05}$ |
| Ours | $\mathbf{0.043 \pm 0.002}$ | $\mathbf{3.77 \pm 0.02}$ | $\mathbf{0.012 \pm 0.001}$ | $\mathbf{3.62 \pm 0.05}$ | $\mathbf{0.071 \pm 0.001}$ | $\mathbf{5.96 \pm 0.01}$ | $0.092 \pm 0.002$ | $8.41 \pm 0.04$ |

Table 3: Model comparisons. MAE ($\downarrow$) and Log-lik (per event) ($\uparrow$) on test data. The first group of methods (FEN, NSPDE, CNN-ODE) contains neural spatiotemporal dynamical models and are evaluated using MAE. The second group (DSTPP, NSTPP, AutoSTPP) contains neural spatiotemporal point process models and are evaluated using Log-lik. Error bars represent one standard error calculated over 5 realizations of a random seed controlling the model initialization.

showed poor results and failed to beat the simple constant intensity baseline, however AutoSTPP shows very strong performance, although still not outperforming our method on the synthetic data, and being close to it on the Scalar Flow dataset.

These experiments show the challenging nature of modeling randomly observed dynamical systems, and demonstrate our method's properties and strong performance.

# 6 RELATED WORK

**Neural Point Processes.** Traditionally, intensity functions are simple parametric functions incorporating domain knowledge about the system being modeled. While simple and interpretable, this approach might require strong domain knowledge and might have limited expressivity due to overly simplistic form of the intensity function. These limitations lead to the development of neural point processes which parameterize the intensity function by a neural network, leading to improved flexibility and expressivity. For example Mei & Eisner (2017); Omi et al. (2019); Jia & Benson (2019); Zuo et al. (2020) use neural networks to model time-dependent intensity functions in a flexible and efficient manner, with neural architectures ranging from recurrent neural networks to transformers and neural ODEs. These techniques consistently outperform classical intensity parameterizations and do not require strong domain expertise to choose the right form of the intensity function. Other works extend this idea to marked point processes where intensity is a function of time and also of a discrete or continuous mark. For example, Chen et al. (2021); Zhu et al. (2021); Zhou et al. (2022); Zhou & Yu (2023); Yuan et al. (2023) use spatial coordinates as a mark and model the intensity using a wide range of methods aimed at improving flexibility and efficiency. Works such as Du et al. (2016); Xiao et al. (2019); Boyd et al. (2020) use discrete observations as marks, and Sharma et al. (2018) use continuous marks with underlying dynamic model, which makes their work most similar to ours, with major differences related to the model, training process, and them working only with temporal processes, whereas we are dealing with more general spatiotemporal processes.

**Neural Spatiotemporal Dynamic Models.** To model the spatiotemporal dynamics our method uses the "encode-process-decode" approach employed in many other works. The main idea of the approach consists of taking a set of initial observations and using it to estimate the latent initial state. A dynamics function is then used to map the latent state forward in time to other time points either discretely (Long et al., 2018; HAN et al., 2022; Wu et al., 2022) or continuously (Yildiz et al., 2019; Salvi et al., 2022). Finally, the latent state is mapped to the observations using discrete (Yildiz et al., 2019; Wu et al., 2022) or continuous (Yin et al., 2023; Chen et al., 2023) decoder. Such methods assume multiple observations per time point and do not model the observation times and locations. However, in contrary to many previous works that use the encode-process-decode approach to learn a discriminative model, we use the auto-encoding variational Bayes to learn a generative model of the underlying dynamical system.

**Implicit Neural Representations.** Our model represents the continuous latent spatiotemporal state $\boldsymbol{u}(t, \boldsymbol{x})$ in terms of a function $\phi(\tilde{\boldsymbol{z}}(t), \boldsymbol{x})$. This approach to representing continuous field is called

implicit neural representations (Park et al., 2019; Chen & Zhang, 2019; Mescheder et al., 2019; Chibane et al., 2020), and it has found applications in many other works related to modeling of spatiotemporal systems (Chen et al., 2023; Yin et al., 2023).

## 7 CONCLUSION

In this work, we developed a method for modeling spatiotemporal dynamical systems from marked point process observations. Along with our method, we proposed an interpolation-based technique to greatly speed up the training time. In the experiments we showed that our method effectively utilizes the context of various lengths to improve accuracy of the initial state inference, uses the system state observations to better model the observation times and locations, and that low-resolution linear interpolation is sufficient to accurately represent the latent trajectories. We further demonstrated that our method achieves strong performance on challenging spatiotemporal datasets and outperforms other methods from the literature.

**Limitations.** Using a Poisson process for sampling event times and locations can be limiting in certain applications. First, the assumption of non-overlapping time points may not hold in scenarios where effectively simultaneous observations occur, such as in high-frequency sensor networks. Second, the lack of interaction between the event occurrences and system dynamics may not hold if the act of observation influences the system (e.g., triggering human actions or affecting sensor availability). Finally, during data collection faithfully capturing the event intensity requires sufficiently dense sensor placement, especially in high-intensity regions. Future work could relax these assumptions to better model such scenarios, for example, by using different point process models and adapting the architecture to efficiently account for potential interactions between the observation events and system dynamics.

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

# A  DATA

As described in Section 3, we assume the following data generating process

$$\boldsymbol{u}(t_1, \cdot) \sim p(\boldsymbol{u}), \quad \frac{\partial \boldsymbol{u}(t, \boldsymbol{x})}{\partial t} = F(\boldsymbol{u}(t, \boldsymbol{x}), \nabla \boldsymbol{u}(t, \boldsymbol{x}))), \tag{20}$$

$$t_i, \boldsymbol{x}_i \sim \mathrm{nhpp}(\lambda(\boldsymbol{u}(t, \boldsymbol{x}))), \qquad\qquad i = 1, \dots, N, \tag{21}$$

$$\boldsymbol{y}_i \sim p(\boldsymbol{y}_i | g(\boldsymbol{u}(t_i, \boldsymbol{x}_i))), \qquad\qquad i = 1, \dots, N, \tag{22}$$

We selected three commonly used PDE systems: Burgers', Shallow Water, and Navier-Stokes with transport. Next we discuss the data generating process for each dataset.

## A.1  BURGERS'

The Burgers' system in 1D is characterized by a 1-dimensional state $u(t, x)$ and the following PDE dynamics:

$$\frac{\partial u}{\partial t} + u \frac{\partial u}{\partial x} = \nu \frac{\partial^2 u}{\partial t^2}, \tag{23}$$

where $\nu$ is the diffusion coefficient. We obtain data for this system from the PDEBench dataset (Takamoto et al., 2023). The system is simulated on a time interval $[0, 2]$ and on a spatial domain $[0, 1]$ with periodic boundary conditions. The dataset contains 10000 trajectories, each trajectory is evaluated on a uniform temporal grid with 101 time points, and uniform spatial grid with 256 nodes.

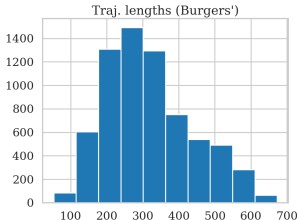

Figure 7: Trajectory lengths for Burgers' dataset.

Since the data is discrete, we define continuous field $u(t, x)$ by linearly interpolating the data across space and time. Next, we use the interpolant to define the intensity function

$$\lambda(u(t, x)) = 1600 |u(t, x)|, \tag{24}$$

and use it to simulate the non-homogeneous Poisson process using the thinning algorithm (Lewis & Shedler, 1979). As the result, for each the 10000 trajectories, we obtain a sequence of time points and spatial locations $\{t_i, x_i\}_{i=1}^N$, where $N$ can be different for each trajectory. Finally, we compute the observations as

$$\boldsymbol{y}_i = u(t_i, x_i). \tag{25}$$

We use the first 0.5 seconds as the context for the initial state inference, and further filter the dataset to ensure that each trajectory has at least 10 points in the context, resulting in approximately 7000 trajectories. We use 80%/10%/10% split for training/validation/testing.

## A.2  SHALLOW WATER

The Shallow Water system in 2D is characterized by a 3-dimensional state

$$\boldsymbol{u}(t, x) = [h(t, \boldsymbol{x}), u(t, \boldsymbol{x}), v(t, \boldsymbol{x})] \tag{26}$$

and the following PDE dynamics:

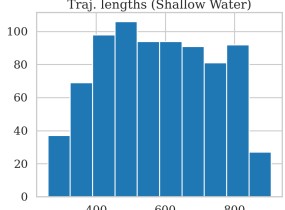

$$\frac{\partial h}{\partial t} + \frac{\partial(hu)}{\partial x} + \frac{\partial(hv)}{\partial y} = 0, \tag{27}$$

$$\frac{\partial u}{\partial t} + u \frac{\partial u}{\partial x} + v \frac{\partial u}{\partial y} + g \frac{\partial h}{\partial x} = 0, \tag{28}$$

$$\frac{\partial v}{\partial t} + u \frac{\partial v}{\partial x} + v \frac{\partial v}{\partial y} + g \frac{\partial h}{\partial y} = 0, \tag{29}$$

Figure 8: Trajectory lengths for Shallow Water dataset.

where $g$ is the gravitational acceleration, $h(t, \boldsymbol{x})$ is wave height, and $u(t, \boldsymbol{x})$ and $v(t, \boldsymbol{x})$ are horizontal and vertical velocities, respectively. We obtain data for this system from the PDEBench dataset

(Takamoto et al., 2023). The system is simulated on a time interval $[0, 1]$ and on a spatial domain $[-2.5, 2.5]^2$. The dataset contains 1000 trajectories, each trajectory is evaluated on a uniform temporal grid with 101 time points, and uniform 128-by-128 spatial grid.

Since the data is discrete, we define continuous field $\boldsymbol{u}(t, x)$ by linearly interpolating the data across space and time. Next, we use the interpolant to define the intensity function

$$\lambda(\boldsymbol{u}(t, \boldsymbol{x})) = 500|h(t, \boldsymbol{x}) - 1|, \tag{30}$$

indicating that measurements are made only when the wave height is below or above the baseline of 1. We use the intensity function to simulate the non-homogeneous Poisson process using the thinning algorithm (Lewis & Shedler, 1979). As the result, for each of the 1000 trajectories, we obtain a sequence of time points and spatial locations $\{t_i, x_i\}_{i=1}^{N}$, where $N$ can be different for each trajectory. Finally, we compute the observations as

$$\boldsymbol{y}_i = h(t_i, \boldsymbol{x}_i), \tag{31}$$

observing only the wave height.

We use the first 0.1 seconds as the context for the initial state inference, and further filter the dataset to ensure that each trajectory has at least 10 points in the context, resulting in approximately 800 trajectories. We use 80%/10%/10% split for training/validation/testing.

## A.3 NAVIER-STOKES

The Navier-Stokes system with a transport equation is characterized by a 4-dimensional state

$$\boldsymbol{u}(t, \boldsymbol{x}) = [c(t, \boldsymbol{x}), u(t, \boldsymbol{x}), v(t, \boldsymbol{x}), p(t, \boldsymbol{x})] \tag{32}$$

and the following PDE dynamics:

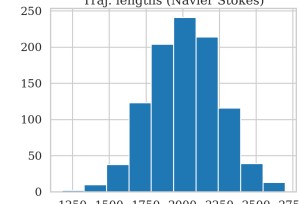

Figure 9: Trajectory lengths for Navier-Stokes dataset.

$$\frac{\partial u}{\partial x} + \frac{\partial v}{\partial y} = 0, \tag{33}$$

$$\frac{\partial u}{\partial t} + u\frac{\partial u}{\partial x} + v\frac{\partial u}{\partial y} = -\frac{\partial p}{\partial x} + \nu\left(\frac{\partial^2 u}{\partial x^2} + \frac{\partial^2 u}{\partial y^2}\right), \tag{34}$$

$$\frac{\partial v}{\partial t} + u\frac{\partial v}{\partial x} + v\frac{\partial v}{\partial y} = -\frac{\partial p}{\partial y} + \nu\left(\frac{\partial^2 v}{\partial x^2} + \frac{\partial^2 v}{\partial y^2}\right) + c, \tag{35}$$

$$\frac{\partial c}{\partial t} = -u\frac{\partial c}{\partial x} - v\frac{\partial c}{\partial y} + \nu\left(\frac{\partial^2 c}{\partial x^2} + \frac{\partial^2 c}{\partial y^2}\right), \tag{36}$$

where the diffusion constant $\nu = 0.002$, and $c(t, \boldsymbol{x})$ is concentration of the transported species, $p(t, \boldsymbol{x})$ is pressure, and $u(t, \boldsymbol{x})$ and $v(t, \boldsymbol{x})$ are horizontal and vertical velocities, respectively. For each trajectory, we start with zero initial velocities and pressure, and the initial scalar field $c_0(x, y)$ is generated as:

$$\tilde{c}_0(x, y) = \sum_{k,l=-N}^{N} \lambda_{kl} \cos(2\pi(kx + ly)) + \gamma_{kl} \sin(2\pi(kx + ly)), \tag{37}$$

$$c_0(x, y) = \frac{\tilde{c}_0(x, y) - \min(\tilde{c}_0)}{\max(\tilde{c}_0) - \min(\tilde{c}_0)}, \tag{38}$$

where $N = 2$ and $\lambda_{kl}, \gamma_{kl} \sim \mathcal{N}(0, 1)$.

We use PhiFlow (Holl et al., 2020) to solve the PDEs. The system is simulated on a time interval $[0, 1]$ and on a spatial domain $[0, 1]^2$ with periodic boundary conditions. The solution is evaluated at randomly selected spatial locations and time points. We use 1089 spatial locations and 25 time points. The spatial and temporal grids are the same for all trajectories. The dataset contains 1000 trajectories.

Since the data is discrete, we define continuous field $\boldsymbol{u}(t, x)$ by linearly interpolating the data across space and time. Next, we use the interpolant to define the intensity function

$$\lambda(\boldsymbol{u}(t, \boldsymbol{x})) = 2000|c(t, \boldsymbol{x})|, \tag{39}$$

indicating that measurement intensity is proportional to the species concentration. We use the intensity function to simulate the non-homogeneous Poisson process using the thinning algorithm (Lewis & Shedler, 1979). As the result, for each of the trajectories, we obtain a sequence of time points and spatial locations $\{t_i, x_i\}_{i=1}^N$, where $N$ can be different for each trajectory. Finally, we compute the observations as

$$\boldsymbol{y}_i = c(t_i, \boldsymbol{x}_i), \tag{40}$$

observing only the species concentrations.

We use the first 0.5 seconds as the context for the initial state inference, and further filter the dataset to ensure that each trajectory has at least 10 points in the context, resulting in all 1000 trajectories satisfying this condition. We use 80%/10%/10% split for training/validation/testing.

### A.4    SCALAR FLOW

To demonstrate capabilities of our model on a real-world process, we use the Scalar Flow Dataset (Eckert et al., 2019). This dataset consists of videos of smoke plumes rising in hot air. Each video has 150 frames recorded over 2.5 seconds, and each frame has the resolution of 1080x1920 pixels. The observations are postprocessed camera images of the smoke plumes taken from multiple views. For simplicity, we use only the front view. We downsample the original frames from 1080x1920 to 120x213 pixels and smooth out the high-frequency noise by applying Gaussian smoothing with the standard deviation of 1.5.

Then, we emulate a real-world measurement process where each "sensor" (pixel) makes a measurement with the probability directly proportional to the observed smoke density (higher density implies more frequent measurements). In particular we set the observation probability to $0.03 \times$ smoke density, resulting in 2000 observations per trajectory on average. As a result, we have a sequence of smoke density observations made at different time points and spatial locations. Note that we do not make any Poisson process assumptions here. Instead, we use a realistic measurement process where sensors are utilized only when some action in the measured process stars to happen, which is the application we are targeting in our work.

We use the first half (1.25 sec.) of each trajectory as the context for the initial state inference. In total, the dataset contains 100 trajectories. We train our and other models on 80 trajectories, and use 10 trajectories for validation, and 10 for testing.

## B    MODEL, POSTERIOR, AND ELBO

### B.1    MODEL

As described in Section 4.1 our model is defined as

$$\boldsymbol{z}_1 \sim p(\boldsymbol{z}_1), \quad \frac{d\boldsymbol{z}(\tau)}{dt} = f(\boldsymbol{z}(\tau)), \tag{41}$$

$$\tilde{\boldsymbol{z}}(t) = \text{interpolate}(t; \boldsymbol{z}(\tau_1), \ldots, \boldsymbol{z}(\tau_n)), \tag{42}$$

$$\boldsymbol{u}(t, x) = \phi(\tilde{\boldsymbol{z}}(t), \boldsymbol{x}), \tag{43}$$

$$t_i, \boldsymbol{x}_i \sim \text{nhpp}(\lambda(\boldsymbol{u}(t, \boldsymbol{x}))), \qquad\qquad i = 1, \ldots, N, \tag{44}$$

$$\boldsymbol{y}_i \sim p(\boldsymbol{y}_i | g(\boldsymbol{u}(t_i, \boldsymbol{x}_i))), \qquad\qquad i = 1, \ldots, N.. \tag{45}$$

The corresponding joint distribution is

$$p(\boldsymbol{z}_1, \{t_i, \boldsymbol{x}_i, \boldsymbol{y}_i\}_{i=1}^N) = p(\boldsymbol{z}_1) p(\{t_i, \boldsymbol{x}_i\}_{i=1}^N | \boldsymbol{z}_1) \prod_{i=1}^N p(\boldsymbol{y}_i | \boldsymbol{z}_1, t_i, \boldsymbol{x}_i), \tag{46}$$

where

$$p(\boldsymbol{z}_1) = \mathcal{N}(\boldsymbol{0}, I), \tag{47}$$

$$p(\{t_i, \boldsymbol{x}_i\}_{i=1}^N | \boldsymbol{z}_1) = \prod_{i=1}^N \lambda(\boldsymbol{u}(t_i, \boldsymbol{x}_i)) \exp\left(-\int \lambda(\boldsymbol{u}(t, \boldsymbol{x})) d\boldsymbol{x} dt\right), \tag{48}$$

$$p(\boldsymbol{y}_i | \boldsymbol{z}_1, t_i, \boldsymbol{x}_i) = \mathcal{N}(g(\boldsymbol{u}(t_i, \boldsymbol{x}_i)), \sigma_{\boldsymbol{y}}^2 I), \tag{49}$$

where $\mathcal{N}$ is the normal distribution, $\boldsymbol{0}$ is a zero vector, and $I$ is the identity matrix. We set $\sigma_{\boldsymbol{y}}$ to $10^{-3}$.

## B.2 POSTERIOR

We define the approximate posterior with $\psi = [\psi_\mu, \psi_{\sigma^2}]$ as

$$q(\boldsymbol{z}_1; \psi) = \mathcal{N}(\boldsymbol{z}_1 | \psi_\mu, \mathrm{diag}(\psi_{\sigma^2})), \tag{50}$$

where $\mathcal{N}$ is the normal distribution, and $\mathrm{diag}(\psi_{\sigma^2})$ is a matrix with vector $\psi_{\sigma^2}$ on the diagonal.

As discussed in Section 4.3, the encoder maps the context $\{t_i, \boldsymbol{x}_i, \boldsymbol{y}_i\}_{i=1}^{N_{\mathrm{ctx}}}$ to the local variational parameters $\psi$ (which we break up into $[\psi_\mu, \psi_{\sigma^2}]$). The encoder first converts the context sequence $\{t_i, \boldsymbol{x}_i, \boldsymbol{y}_i\}_{i=1}^{N_{\mathrm{ctx}}}$ to a sequence of vectors $\{\boldsymbol{b}_i\}_{i=1}^{N_{\mathrm{ctx}}}$ and then adds an aggregation token, which gives the following input sequence: $\{\{\boldsymbol{b}_i\}_{i=1}^{N_{\mathrm{ctx}}}, [\mathrm{AGG}]\}$. We pass it though the transformer encoder and read the value of the aggregation token at the last layer, which we denote by $\overline{[\mathrm{AGG}]}$. Then, we map $\overline{[\mathrm{AGG}]}$ to $[\psi_\mu, \psi_{\sigma^2}]$ as follows:

$$\psi_\mu = \mathrm{Linear}(\overline{[\mathrm{AGG}]}), \tag{51}$$

$$\psi_{\sigma^2} = \exp\left(\mathrm{Linear}(\overline{[\mathrm{AGG}]})\right), \tag{52}$$

where Linear are separate linear mappings.

## B.3 ELBO

Given definitions in the previous sections, we can write the ELBO as

$$\mathcal{L} = \int q(\boldsymbol{z}_1) \ln \frac{p(\boldsymbol{z}_1, \{t_i, \boldsymbol{x}_i, \boldsymbol{y}_i\}_{i=1}^N)}{q(\boldsymbol{z}_1)} d\boldsymbol{z}_1 \tag{53}$$

$$= \int q(\boldsymbol{z}_1) \ln \frac{\prod_{i=1}^N p(\boldsymbol{y}_i | \boldsymbol{z}_1, t_i, \boldsymbol{x}_i) p(\{t_i, \boldsymbol{x}_i\}_{i=1}^N | \boldsymbol{z}_1) p(\boldsymbol{z}_1)}{q(\boldsymbol{z}_1)} d\boldsymbol{z}_1 \tag{54}$$

$$= \int q(\boldsymbol{z}_1) \ln \prod_{i=1}^N p(\boldsymbol{y}_i | \boldsymbol{z}_1, t_i, \boldsymbol{x}_i) d\boldsymbol{z}_1 \tag{55}$$

$$+ \int q(\boldsymbol{z}_1) \ln p(\{t_i, \boldsymbol{x}_i\}_{i=1}^N | \boldsymbol{z}_1) d\boldsymbol{z}_1 \tag{56}$$

$$+ \int q(\boldsymbol{z}_1) \ln \frac{p(\boldsymbol{z}_1)}{q(\boldsymbol{z}_1)} d\boldsymbol{z}_1 \tag{57}$$

$$= \underbrace{\sum_{i=1}^N \mathbb{E}_{q(\boldsymbol{z}_1)} \left[\ln p(\boldsymbol{y}_i | \boldsymbol{z}_1, t_i, \boldsymbol{x}_i)\right]}_{\mathcal{L}_1} \tag{58}$$

$$+ \underbrace{\mathbb{E}_{q(\boldsymbol{z}_1)} \left[\sum_{i=1}^N \ln \lambda(\boldsymbol{u}(t_i, \boldsymbol{x}_i)) - \int \lambda(\boldsymbol{u}(t, \boldsymbol{x})) d\boldsymbol{x} dt\right]}_{\mathcal{L}_2} \tag{59}$$

$$- \underbrace{\mathrm{KL}[q(\boldsymbol{z}_1) \| p(\boldsymbol{z}_1)]}_{\mathcal{L}_3}. \tag{60}$$

$$= \mathcal{L}_1 + \mathcal{L}_2 - \mathcal{L}_3. \tag{61}$$

**Computing ELBO.** We compute the ELBO as follows:

1. Compute local variational parameter from the context as $\psi_\mu, \psi_{\sigma^2} = \mathrm{Encoder}(\{t_i, \boldsymbol{x}_i, \boldsymbol{y}_i\}_{i=1}^{N_{\mathrm{ctx}}})$

2. Sample the latent initial state as $\boldsymbol{z}_1 \sim \mathcal{N}(\boldsymbol{z}_1 | \psi_\mu, \mathrm{diag}(\psi_{\sigma^2}))$ using reparameterization

3. Evaluate $\boldsymbol{z}(\tau_1), \ldots, \boldsymbol{z}(\tau_n)$ by solving $\frac{d\boldsymbol{z}(\tau)}{dt} = f(\boldsymbol{z}(\tau))$ with $\boldsymbol{z}_1$ as the initial condition. We solve the ODE using `torchdiffeq` package (Chen et al., 2018).

4. Compute the latent states $\tilde{\boldsymbol{z}}(t_1), \ldots, \tilde{\boldsymbol{z}}(t_N)$ via interpolation as
   $\tilde{\boldsymbol{z}}(t) = \text{interpolate}(t; \boldsymbol{z}(\tau_1), \ldots, \boldsymbol{z}(\tau_n))$

5. Compute $\boldsymbol{u}(t_1, \boldsymbol{x}_1), \ldots, \boldsymbol{u}(t_N, \boldsymbol{x}_N)$ as $\boldsymbol{u}(t_i, \boldsymbol{x}_i) = \phi(\tilde{\boldsymbol{z}}(t_i), \boldsymbol{x}_i)$

6. Compute $\mathcal{L}_1$ using Monte Carlo integration

7. Compute $\mathcal{L}_2$ using Monte Carlo integration, with the intensity integral computed using `torchquad` (Gómez et al., 2024) package.

8. Compute $\mathcal{L}_3$ analytically

Sampling is done using reparametrization (Kingma & Welling, 2013) to allow for exact gradient evaluation via backpropagation. Monte Carlo integration is done using sample size of one.

## C Experiments Setup

### C.1 Datasets

See Appendix A for all details about dataset generation. For all datasets we use 80%/10%10% train/validation/test splits. The context size $N_{\text{ctx}}$ is determined for each trajectory separately based on the time interval that we consider to be the context, for Burgers' and Navier-Stokes it is the first 0.5 seconds (out of 2 seconds), while for Shallow Water it is 0.1 seconds (out of 1 second). Since the number of events occurring within this fixed time interval is different for each trajectory, the context size consequently differs across trajectories. See encoder architecture description below for details on how different context lengths are processed.

### C.2 Training, Validation, and Testing

We use AdamW (Loshchilov & Hutter, 2019) optimizer with constant learning rate 3e-4 (we use linear warmup for first 250 iterations). We train for 25000 iterations, sampling a random minibatch from the training dataset at every iteration. We do not use any kind of scaling for the ELBO terms.

We compute validation error (MAE on observations) on a single random minibatch from the validation set every 125 iterations. We save the current model weights if the validation error averaged over last 10 validation runs is the smallest so far.

To simulate the model's dynamics we use differentiable ODE solvers from `torchdiffeq` package (Chen et al., 2018). In particular, we use the dopri5 solver with rtol = atol = $10^{-5}$ without the adjoint method. To evaluate the intensity integral $\int \lambda(\boldsymbol{u}(t, \boldsymbol{x}) d\boldsymbol{x} dt$ we use Monte Carlo integration from the `torchquad` (Gómez et al., 2024) package with 32 randomly sampled points for training, and 256 randomly sampled points for testing, we found these sample sizes to be sufficient for producing small variance of the integral estimation in our case.

### C.3 Encoder and Model Architectures

**Encoder.** We use padding to efficiently handle input contexts of varying length. The time points, coordinates, and observations are linearly projected to the embedding space (128-dim for Burgers' and Shallow Water, and 192-dim for Navier-Stokes). Then, the projections are summed and passed through a stack of transformer encoder layers (4 layers for Burgers' and Shallow Water, and 5 layers for Navier-Stokes, 4 attention heads were used in all cases). We introduce a learnable aggregation token which we append at the end of each input sequence. The final layer's output is read from the last token of the output sequence (corresponds to the aggregation token) and is mapped to the local variational parameters as discussed in Appendix B.

**Model.** The dynamics function $f(\boldsymbol{z}(t))$ is an MLP with 3 layers 368 neurons each, and GeLU (Hendrycks & Gimpel, 2016) nonlinearities. We set resolution of the uniform temporal grid $\tau_1, \ldots, \tau_n$ to $n = 50$. The dynamics function maps the current latent state $\boldsymbol{z}(t)$ to its time derivative $\frac{d\boldsymbol{z}(t)}{dt}$, both are $d_{\boldsymbol{z}}$-dimensional vectors.

The function $\phi(\tilde{\boldsymbol{z}}(t), \boldsymbol{x})$ is an MLP with 3 hidden layers with width 368 for Burgers', 256 for Shallow Water, and 512 for Navier-Stokes. We use GeLU (Hendrycks & Gimpel, 2016) nonlinearities. As the

input the MLP takes the sum of $\tilde{z}(t) \in \mathbb{R}^{d_z}$ with $\mathrm{proj}(x) \in \mathbb{R}^{d_z}$, where $\mathrm{proj}$ is a linear projection, and maps the sum to the output $u(t, x) \in \mathbb{R}^{d_u}$.

The intensity function $\lambda(u(t, x))$ is an MLP with 3 layers, 256 neurons each, and GeLU (Hendrycks & Gimpel, 2016) nonlinearities. We further exponentiate the output of the MLP and add a small constant (1e-4) to it to ensure the intensity is positive and to avoid numerical instabilities.

We set the observation function $g(u(t, x))$ to the identity function, thus $g(u(t, x)) = u(t, x)$.

## C.4 Model Comparison

### C.4.1 Dynamic Spatiotemporal Models

In this section we discuss neural spatiotemporal models we used for the comparisons. But first, we discuss data preprocessing that we do in order to make these models applicable.

**Data Preprocessing.** All models in this section assume that dense observations are available at every time point, and some further assume the observations are located on a uniform spatial grid. To make our data satisfy these requirements we use time binning and spatial interpolation. First, we use time binning and divide the original time grid into 10 bins and group all time points into 10 groups based on which bin they belong to. We used 10 time bins as it ensured that there was a sufficient number of time points in each bin while also being sufficient to capture changes of the system state. We denote by $t_i$ the time point that represents the bin $i$. Next, we use all observations that belong to bin $i$ and interpolate them to a uniform 32x32 spatial grid. In many cases some points of the spatial grid were outside of the convex hull of the interpolation points, so we set values of such points to -1 (values of the state range from 0 to 1). We used linear interpolation. We denote the resulting interpolated values at time $t_i$ as $u_i$ which is the interpolated system state at the 32x32 spatial grid. When we compute the loss between $u_i$ and the corresponding model prediction we use only those spatial locations that had proper interpolation values (i.e. were inside the convex hull of the interpolation points).

**Median predictor.** This is a simple baseline where we compute the median of the observations $y_i$ on the training data and use it as the prediction on the test data.

**CNN-ODE.** This is another relatively simple baseline based on latent neural ODEs (Chen et al., 2018). It operates in some sense similarly to our model as it takes a sequence of initial observations (context) $y_1, \ldots, y_{N_{\mathrm{ctx}}}$ and uses a CNN encoder to map the context to the latent initial state $z_1$. Then it maps $z_1$ to $z_2, \ldots, z_N$ via a latent ODE, and finally decodes $z_i$ to obtain $u_i$ via a CNN decoder. In our case we used the first two observations $u_1$ and $u_2$ as the context.

**FEN.** This model discretizes the spatial domain into a mesh and uses a graph neural network to model the dynamics of the state at each node of the mesh. The model starts at $u_1$ and maps it directly to $u_2, \ldots, u_N$ via an ODE solver without using any context. We simulated the dynamics as stationary and autonomous, with the free-form term active. The dynamics function is a 2-layer MLP with the width of 512 neurons and ReLU nonlinearities. All other hyper-parameters were left as defaults as we found that changing them did not improve the results. We used the official code from Lienen & Günnemann (2022).

**NSPDE.** This model applies a point-wise transformation to map the system state at every spatial node to a high-dimensional latent representation, then uses the fixed-point method to simulate the dynamics at every node, and finally maps the high-dimensional latent states back to the observation space to obtain the predictions. We used the fixed-point method with maximum number of available modes and a single iteration as we found that larger number of iterations does not improve the results. All other hyper-parameters were left as defaults as we found that changing them did not improve the results. We used the official code from Salvi et al. (2022).

### C.4.2 Neural Spatiotemporal Processes

**Constant Intensity Baseline.** As a simple baseline we use the following intensity function $\lambda(t, x) = c$, where $c$ is a learnable parameter that we fit on the training data.

| $d_{\mathbf{z}}$ | $h_\phi$ | $h_f$ | | | |
|---|---|---|---|---|---|
| | | 23 | 92 | 368 | 512 |
| 40 | 128 | 0.090 | 0.091 | 0.084 | 0.081 |
| 40 | 512 | 0.075 | 0.073 | 0.074 | 0.072 |
| 368 | 128 | 0.084 | 0.083 | 0.084 | 0.082 |
| 368 | 512 | 0.072 | 0.073 | 0.073 | 0.072 |

Table 4: Dependence of the test MAE on latent space dimension $d_{\mathbf{z}}$, dynamics complexity $h_f$ and decoder complexity $h_\phi$.

**DSTPP.**  This model uses a transformer-based encoder to condition a diffusion-based density model on the event history. We use 500 time steps and 500 sampling steps, and train for at most 12 hours. All other hyper-parameters were left as defaults as we found that changing them did not improve the results or made the training too slow. We used the official code from Yuan et al. (2023).

**NSTPP.**  This model represents the vent history in terms of a latent state governed by a neural ODE, with the latent state being discontinuously updated after each event. The latent state is further used to represent the distribution of spatial locations via a normalizing flow model. We used the attentive CNF version of the model. All other hyper-parameters were left as defaults as we found that changing them did not improve the results. We used the official code from Chen et al. (2021).

**AutoSTPP.**  This model uses dual network approach for flexible and efficient modeling of spatiotemporal processes. We used 5 product networks with 2 layers 128 neurons each and hyperbolic tangent nonlinearities. The step size was set to 20 and number of steps was set to 101 in each direction. All other hyper-parameters were left as defaults as we found that changing them did not improve the results. We used the official code from Zhou & Yu (2023).

## D  EFFECTS OF DYNAMICS FUNCTION AND DECODER COMPLEXITY

Our model (similarly to previous competing models) consists of a composition of mappings that are parameterized by deep neural networks. Therefore, the role and importance of different model components depends on a number of hyperparameters. This, in turn, is related to the result reported in the main text (Figure 5), where we observed that, in the case of linear interpolation, decreasing the number of interpolation points did not decrease predictive performance of our model. Decreasing the number of interpolation points to $n = 2$ corresponds to replacing non-linear dynamics function with a linear model, but this decrease in complexity of the dynamics model can, in some cases, be compensated by the capacity of decoder function. In this section, we conduct a series of experiments to better understand the role of the dynamics and decoder functions in our model. In particular, we look at the model's predictive accuracy as a function of the latent space dimension $d_{\mathbf{z}}$, and complexity of the dynamics function $f$ and decoder $\phi$. Since both $f$ and $\phi$ are 3-layer MLPs, we define their complexities as the width of the hidden layers $h_f$ and $h_\phi$, respectively. In particular, we look at low- and high-dimensional latent spaces ($d_{\mathbf{z}} = 40$ and $d_{\mathbf{z}} = 368$); we consider medium and large decoder function $\phi$ (with $h_\phi = 128$ and $h_\phi = 512$), and dynamics functions of different complexities ranging from $h_f = 23$ to $h_f = 512$. We use linear interpolation with 50 interpolation points to allow the model represent complex latent trajectories if it helps to fit the data better. We use the Navier-Stokes data (since this is the most complex synthetic dataset in our work) and report only the mean absolute error (as log-likelihood follows the same pattern). The results are in Table 4.

As can be seen, for the low-dimensional latent space ($d_{\mathbf{z}} = 40$) and medium-sized decoder (with $h_\phi = 128$) increasing complexity of the dynamics function $h_f$ leads to a meaningful reduction of MAE. However, after increasing complexity of the decoder from $h_\phi = 128$ to $h_\phi = 512$, increasing $h_f$ results in no significant improvements. Then, if we switch to a high-dimensional latent space ($d_{\mathbf{z}} = 368$), we see that complexity of the decoder $h_\phi$ becomes the main determinant of the model's predictive accuracy, with dynamics function complexity $h_f$ having no observable effect.

In summary, we see that increasing complexity of the dynamics function of the neural ODE can be useful in some cases (low-dimensional latent space and medium-sized decoder), but with a sufficiently

large decoder and large latent spaces its benefits become less apparent, meaning that in such cases the dynamics can be modeled using simple function approximators. While the experimental results show that in some cases accurate predictions can be achieved with simple dynamics, enabling the model to represent complex dynamics via Neural ODEs can be useful also for cases where complex latent trajectories are expected or enforced.

