# OpenReview forum: "Learning Spatiotemporal Dynamical Systems from Point Process Observations"
_ICLR.cc/2025/Conference — ICLR 2025 Spotlight_

### Official Review · Reviewer_Nyvc · 2024-10-29

**Soundness:** 3
**Presentation:** 3
**Contribution:** 3
**Rating:** 8
**Confidence:** 2

**Summary:**

This paper presents a novel method for modeling spatiotemporal dynamical systems from point process observations. The model integrates techniques from neural differential equations, neural point processes, implicit neural representations, and amortized variational inference. The authors also introduce a technique to speed training by addressing a computational bottleneck in latent state evaluation. The experimental results demonstrate the effectiveness of the model on challenging spatiotemporal datasets.

**Strengths:**

- The paper is well-written and technically sound. The methodology is clearly presented, and the experimental setup is detailed
- The proposed model is technically sound. It effectively combines techniques from various fields, including neural differential equations, neural point processes and amortized variational inference
- experiments and "ablations studies" are comprehensive, showing the impact of many parameters of the model

**Weaknesses:**

- While focusing on predictive capability and computational efficiency, discussing the interpretability of the model would enhance its value. Can something be said about the dynamical system?
- A little more discussion around the limitation of the Poisson process, and potential solution would have been welcome.

**Questions:**

Questions are related to the weaknesses:
Could you address the issue of interpretability and the Poisson process a bit more

---

> ### Author Response · Authors · 2024-11-21
>
> **Q1:** While focusing on predictive capability and computational efficiency, discussing the interpretability of the model would enhance its value. Can something be said about the dynamical system?
> **A1:** In Section 4.1, we describe the structure and function of each component of our model, which makes it easier to understand and interpret. However, interpreting the meaning and structure of the learned latent space and dynamics is more difficult. Indeed, deep generative models, despite their flexibility and strong predictive performance, are generally hard to interpret. However, this is a limitation of deep generative models in general, and not something specific to our approach. Interpretability of deep generative models would require a separate study, which is outside the scope of our present work.
>
> -------------------------------
>
> **Q2:** A little more discussion around the limitation of the Poisson process, and potential solution would have been welcome.
> **A2:** In line 497 we discussed what we believe are the main limitations of using the Poisson process. These limitations could be alleviated by using different point process models and adapting the architecture to efficiently account for potential interactions between the observation events and system dynamics. We will extend the discussion on limitations but leave the extensions to be considered in future work where such modeling assumptions are needed.
>
> We will extend the discussion of these two topics and add it to the revised manuscript.

---

### Official Review · Reviewer_BCH2 · 2024-10-31

**Soundness:** 4
**Presentation:** 3
**Contribution:** 3
**Rating:** 8
**Confidence:** 4

**Summary:**

A composition of different ML methods is presented to simulate spatiotemporal processes from point observations without access to the whole spatial field. The proposed approach encodes sparse context point observations into a latent state. In this latent state, the dynamic process evolution is integrated over large time steps, while a fine temporal resolution is obtained via interpolation in the latent state. A decoder projects the predicted latent states back to the observation space.

**Strengths:**

_Originality:_ The combination of various different branches from machine learning is original. They are composed in an elegant and versatile way to solve spatiotemporal problems efficiently. The use of latent states enforces abstractions that push for generalizability. Intriguingly, the introduced method does not rely on complex architectures, but mostly emerges out of various MLPs that are well placed and wired.

_Quality:_ Claims are well supported with experimental results, which are contrasted against several recent  and competitive ML architectures. Figures are well designed and support the message conveyed in the manuscript.

_Clarity:_ The manuscript is well organized, structured, and written. A rich appendix provides details about the model design, yet a supplementary material to validate the results is missing.

_Significance:_ Results appear significant in terms of how sparse the data is sampled. Three synthetic problems of varying difficulty, as well as a real-world problem demonstrate the applicability of the method. Results are reported in two metrics along with standard deviations, which helps assessing the quality of the forecasts.

**Weaknesses:**

1. Observation function is constrained to a normal distribution with fixed variance. It would be helpful to add arguments of this design choice, to what extend it limits the expressivity of the model, as well as for what problems this formulation is sufficient.
2. Ablations showing the performance under different spatial and temporal sparsities would be highly informative to understand the quality and limitations of the model at different tasks. Presumably, e.g., Navier-Stokes likely depends on more dense samples compared to Shallow Water. Extending this ablation to the other benchmarked methods would also provide valuable insights about the models' data efficacy.
3. Limitations are not reported. It would be valuable to understand the limits of the method, its computational cost, and the time this architecture needs to train. Also, it is unclear to which extend the method can directly be applied to a task at hand or how much fine tuning is involved.
4. No runtime comparison of the different models provided. If I'm not mistaken, the model must be called for each spatial position of interest in each time step, which amounts to a large number of model calls. Thus, to extend on Table 1, please provide information about the runtime of the entire model when generating a rollout of a spatiotemporal sequence of frames.
5. More details about the differences between the introduced method and AutoSTPP would be valuable, given that these two approaches perform almost equally well. For what reason is your method superior to AutoSTPP?

_Minor Comments_
- Typo in line 306, "withing"
- $N_{\text{ctx}}$ is unclear in Figure 4. What value does the variable take? Would be good to have the actual value. EDIT: C.1 provides this information; I thus suggest to refer to C.1 in the Caption of Figure 4.

**Questions:**

1. For what reason is the distribution of the next sensor signal's location predicted? What is the benefit of such a prediction and what computational cost does it impose? If I understand correctly, Table 2 suggests removing the point process model (which simulates the next sensor signal position and time, if I'm correct). At least according to a minimal model when following Occams Razor.
2. The interpolation ablation is very illustrative. Have you tried higher-order interpolations to infer $\hat{z}(t_i)$, i.e., quadratic, cubic? What is the error that incurs from the interpolation compared to modeling the full temporal grid $t_1, \dots, t_N$? Table 1 demonstrates the time improvement when using interpolations; it would be great to see the error associated with the two techniques (Interp. vs Seq.).
3. Have you explored other solvers beyond dopri5, such as Euler, which is much cheaper? Or does the method depend on an adaptive solver to account for potentially different deltas between time steps? Figure 2 somehow suggest that the effectively processed time steps $\tau_m$ are separated by a constant time delta. Is this a requirement of the architecture?
4. How does the latent space dimensionality $d_z$ affect the runtime? Might be interesting to report along with its effect on the parameter count around line 375.

---

> ### Author Response · Authors · 2024-11-21
>
> **Q1:** Observation function is constrained to a normal distribution with fixed variance...
> **A1:** Our model supports and can be easily extended to other observation likelihood models. This would only require a reparameterization such that the observation function $g$ (in Eq. 11) returns parameters of the new distribution.
>
> --------------------
>
> **Q2:** It would be valuable to understand the limits of the method, its computational cost, and the time this architecture needs to train...
> **A2:** As we mention in line 318, our method requires at most 1.5 hours of training. We also discuss limitations of our model at the end of Section 7. Our method, as many others, requires hyperparameter tuning to achieve best predictive performance, but it is rather robust to many hyperparameters, with the latent state dimension $d_z$, and complexities of the dynamics function $f$ and latent state decoder $\phi$ being most important hyperparameters to tune.
>
> --------------------
>
> **Q3:** No runtime comparison of the different models provided ... Please provide information about the runtime of the entire model when generating a rollout of a spatiotemporal sequence of frames.
> **A3:** In our experiments, system dynamics simulation is the largest contributor to the entire model runtime. For example, on the Navier-Stokes dataset, which has the largest number of positions of interest, the runtime split across the model components is: encoder - 10\%, dynamics - 63\%, decoder - 27\%. Note that these numbers are with the latent space interpolation. Without it, dynamics simulation effectively takes all of the model runtime.
>
> --------------------
>
> **Q4:** More details about the differences between the introduced method and AutoSTPP would be valuable...
> **A4:** The major advantage of our model over AutoSTPP is that our model leverages the observed system states $y$ to better model the observation process of $t$ and $x$, while AutoSTPP doesn't.
>
> --------------------
>
> **Q5:** For what reason is the distribution of the next sensor signal's location predicted? What is the benefit of such a prediction and what computational cost does it impose?
> **A5:** Modeling observation times and locations is fundamental for spatio-temporal point processes, which our method combines with simultaneously modeling the observations of the underlying spatio-temporal dynamics. Table 2 indeed shows that if the sensor locations are known, modeling them does not improve the model's predictive accuracy. However, systems we consider in our work produce observations at random spatiotemporal locations, so the sensor locations are unknown at test time. This means we need to predict where the observations will be made. Modeling sensor locations takes ≈15\% of the total runtime.
>
> --------------------
>
> **Q6:** Have you tried higher-order interpolations? What is the error that incurs from the interpolation compared to modeling the full temporal grid?
> **A6:** We didn't consider higher order methods because the nearest neighbor and linear interpolation approach the full temporal grid errors for sufficient number of interpolation points ($n=50$ in our case).
>
> --------------------
>
> **Q7:** Have you explored other solvers beyond dopri5, such as Euler, which is much cheaper? ... Figure 2 somehow suggest that the effectively processed time steps $\tau_m$ are separated by a constant time delta. Is this a requirement of the architecture?
> **A7:** Yes, the time steps $\tau_1, \dots, \tau_n$ are separated by a constant time, which allows to use e.g., the Euler solver. However, our method is not constrained by this assumption and works with both regular and irregular time grids as well as fixed and adaptive step sizes. We tried both Euler and dopri5 solvers with 20 interpolation points ($n=20$), and the training time difference was small (41 min for Euler, and 49 min for dopri5); the predictive performance for both solvers was similar as well (within 3\% of each other).
>
> --------------------
>
> **Q8:** How does the latent space dimensionality affect the runtime? Might be interesting to report along with its effect on the parameter count around line 375.
> **A8:** To answer this question we measure the runtime using on our Navier-Stokes dataset as this is the largest dataset in our work and we use the model with the largest number of parameters for it. We measure the total training time and the total number of model parameters as a function of the latent space dimension $d_z$:
>
> | $d_z$  | Time   | Parameters  |
> |-----|--------|---------|
> | 368 (default) | 45 min | 3022210 |
> | 184 | 43 min | 2721002 |
> | 92  | 43 min | 2570398 |
> | 46  | 43 min | 2495096 |
>
> We see that the both the training time and number of parameters are rather weakly affected by the latent space dimension.
>
> We thank the reviewer for careful reading and comments. We will incorporate the above comments and clarifications into the revised manuscript.

---

### Official Review · Reviewer_f2Uy · 2024-11-04

**Soundness:** 3
**Presentation:** 2
**Contribution:** 3
**Rating:** 6
**Confidence:** 3

**Summary:**

This article is devoted to the problem of learning spatiotemporal dynamics from randomly sampled points in space and time. This problem is particularly well suited for the situation where we have sensors that record a system, and we have to predict also the behavior of the sensors during the dynamics (e.g. meteorological sensors that are carried by currents). The method proposed in this article is based on the use of neural ODEs in a learned latent space.

**Strengths:**

-- The problem of learning spatiotemporal point processes is rather important, and any contribution to this problem should be well welcomed by the scientific community.
-- The overall idea of the article is meaningful.
-- The numerical results are rather good.

**Weaknesses:**

-- Some explanations are not properly given. For instance, I assume that they are using an ODE solver in a latent space because a direct approach would immediately incur into stiffness problems. Why not using a neural PDE solver? Why is it better to learn a latent space and use an ODE solver for a problem that is formulated as a PDE (as in Eqn 5 of the paper)? This is unclear.
-- The latent approach makes the approach less clear, and more out of the control of the user. I suppose the authors have no idea why the encoder creates a certain latent space rather than another. A theoretical approach seems very complicated, in fact the authors limit themselves mostly to empirical results.
-- It is unclear if a general system can be learned in this way. In a sense, we might think of the encoded latent space as a low-degree approximation of the system, but it might be that certain PDE models stemming from Eqn 5 might not be suitably tackled by such approach.
-- One of the main claims is that the model is continuous. An interpolation task should be performed in this case to show that they can handle continuity well. They use interpolation in the method, but it is unclear if in an experiment where portions of the trajectories are completely hidden during training, could be recovered during evaluation.

**Questions:**

My main questions relate the points raised in the weaknesses above.

---

> ### Author Response · Authors · 2024-11-21
>
> **Q1:** Why not using a neural PDE solver? Why is it better to learn a latent space and use an ODE solver for a problem that is formulated as a PDE (as in Eqn 5 of the paper)?
> **A1:** As we describe on line 184, we use a low-dimensional latent state because it allows to simulate the dynamics considerably faster than a full-grid spatiotemporal discretization.
>
> --------------------------
>
> **Q2:** The latent approach makes the approach less clear, and more out of the control of the user. I suppose the authors have no idea why the encoder creates a certain latent space rather than another.
> **A2:** Indeed, deep generative models, despite their flexibility and strong predictive performance, are generally hard to interpret. However, this is a limitation of deep generative models in general, and not something specific to our approach. We also note that, if needed, various model properties might be enforced, for example with penalty losses, architectural choices or even expert-defined parametric model components, thus giving a degree of control over the model and its interpretability.
>
> --------------------------
>
> **Q3:** This encoder-based latent space modeling approach might not be able to model general PDE systems.
> **A3:** Following previous studies, we addressed this concern by testing our model on a wide range of challenging PDE systems, where it showed strong predictive performance. While this is not a proof that our model works for all conceivable PDE systems, this a strong evidence that it works for a wide range of realistic and practical scenarios.
>
> --------------------------
>
> **Q4:** Why is the model claimed to be continuous?
> **A4:** Our model is continuous because it defines the system state at any arbitrary spatial and temporal location, rather than restricting it to predefined discrete grids. We note that all differential equation systems (except selected toy examples that allow closed-form solutions) need some numerical solution methods, such as the adaptive dopri5 solver that we used in our study.

---

### Official Review · Reviewer_UEak · 2024-11-04

**Soundness:** 2
**Presentation:** 4
**Contribution:** 3
**Rating:** 8
**Confidence:** 4

**Summary:**

This is an engineering oriented work that model STPP with intensity driven by a continuous latent states governed by a Neural-ODE, with initial states generated by a transformer encoder. The formulation sounds valid and the proposed benefits are for sparsely distributed data. The main contributions are the new formulation and the interpolation-based speedup technique.

**Strengths:**

- The challenge seems well grounded as the sparse data over a large spatial domain is common for many types of problems, e.g., few agent trajectories over a large geographical domain.
- The method looks (possibly) scalable with low-resolution linear interpolation.
- The math formulation is clear and the empirical results are fair.
- A lot of ablation study, accounting for context size, resolution, removal of components

**Weaknesses:**

* I don't think the paper really answer the question of why it work on sparse data. There is no theoretical analysis / visualization of how the low-dimensional latent space captures the full dynamics from sparse observations. No discussion of information-theoretic bounds on what can be learned from sparse observations. It is reasonable to expect normalizing-flow based method (like Neural-STPP) not working well because the distribution is too localized, but I don't see why your method have an theoretical advantage over SOTA with kernel-based or closed spatial distribution.

**Questions:**

* Can you give me a possible explanation of why it works?
* There is no ablation study on why transformer are used for generating the initial states. Or do you have evidence the initial state part is robust to architecture choice?
* Despite the proposed speedup method, I believe neural-ODE is still untolerably slow and does not scale well. Do you have actual training/ inference time comparison?

---

> ### Author Response · Authors · 2024-11-21
>
> **Q1:** Can you give me a possible explanation of why it works [on sparse data]?
> **A1:** Our model was designed to work on sparse data by 1) modeling the observation times t and locations x via a point process (Eq. 10), and 2) using the observed states y to improve the modeling of observation events (Eq.11). These features allow us to handle sparse observations made at arbitrary spatiotemporal locations, which is in contrary to all previous methods. To the best of our knowledge, previous methods can model only observation times and locations using a point process, or they can model spatio-temporal dynamics assuming that the observations are available at pre-defined time points or dense spatial locations, but but not both.
>
> ----------------
>
> **Q2:** There is no ablation study on why transformer are used for generating the initial states. Or do you have evidence the initial state part is robust to architecture choice?
> **A2:** Transformer was used as a flexible general-purpose architecture supporting training-time parallelization. From our experience, RNN-based encoders could give results on-par with Transformer, but require longer training times.
>
> ----------------
>
> **Q3:** Despite the proposed speedup method, I believe neural-ODE is still untolerably slow and does not scale well. Do you have actual training/ inference time comparison?
> **A3:** To answer this question we compared the training times with neural ODE and a map-based dynamics (where the next state $z_{i+1}$ is evaluated as $z_{i+1} = f(z_i)$). We used the Navier-Stokes dataset as this is the largest of our datasets. As the ODE solver we used dopri5 with high absolute and relative tolerances of $10^{-5}$, which ensured accurate solutions. As the result, training with neural ODE dynamics took 45 minutes, while using map-based dynamics took 40 minutes. We also measured training times with lower absolute and relative tolerances set to $10^{-3}$, which resulted in 42 minutes-long training. So, we see that while using neural ODEs as the dynamics model increases the training time, the increase is quite modest.

---

> > ### Comment · Reviewer_UEak · 2024-12-02
> >
> > Q1: I see your argument. So the definition of "sparsity" in your claim is a synonym of "asynchronous observations and events" and are not parallel to your "sensor network" claim.
> >
> > I would recommend you clarify that because generally people would believe sparsity refers to density / intensity equals zero for most of the places in the spatio-temporal domain, or at least mean large amount of missing events (most events have label 0 instead of 1). Your usage of "sparsity" is quite misleading, although the number of observations could be much fewer than events, there is no lots of zero-values involved. As this is not claimed, you are good.
> >
> > Q2: Sounds good. This is consistent with my general intuition.
> >
> > Q3: I would be excited to see your supplemental material to validate this.
> >
> > I would remain the score.

---

### Meta-Review · Area_Chair_k69c · 2024-12-23

**Metareview:**

The paper proposes a method for learning to model spatio-temporal processes from data that is irregularly sampled in both the spatial and temporal dimensions. It employs an encode-process-decode framework and integrates the following components: point processes (non-homogeneous Poisson processes) for capturing irregular information and dynamics in a latent space, a neural ODE for time-stepping in the latent parameter space, variational inference for learning the parameters of the posterior distribution for the initial latent state, and Implicit neural representations for decoding. The model is evaluated on four datasets and compared against various baselines.

All reviewers acknowledge the novelty of the contribution, which combines existing components in an innovative way. They consider that the authors' claims are well-supported by the experiments and ablation studies. I recommend acceptance.

**Additional Comments On Reviewer Discussion:**

There were limited discussions during the rebuttal, but all reviewers agreed on acceptance.

---

### Decision · Program_Chairs · 2025-01-22

Accept (Spotlight)